# Pre and Post Water Level Behaviour in Punjab: Impact Analysis with DiD Approach

**Yogita Sharma** [1], **Baljinder Kaur Sidana** [2,*], **Sunny Kumar** [2], **Samanpreet Kaur** [3], **Milkho Kaur Sekhon** [2], **Amrit Kaur Mahal** [4] **and Sushant Mehan** [5]

1   Department of Agricultural Economics, Bihar Agricultural University, Bhagalpur 813210, Bihar, India
2   Department of Economics and Sociology, Punjab Agricultural University, Ludhiana 141004, Punjab, India
3   Department of Soil and Water Engineering, Punjab Agricultural University, Ludhiana 141004, Punjab, India
4   Department of Mathematics, Statistics and Physics, Punjab Agricultural University, Ludhiana 141004, Punjab, India
5   Colorado Water Center, Water Management and Systems Research Unit, United States Department of Agriculture–Agricultural Research Service, Fort Collins, CO 80521, USA
*   Correspondence: baljindersidana@pau.edu

**Abstract:** Punjab Agriculture is trapped in the complex nexus of groundwater depletion and food insecurity. The policymakers are concerned about reducing groundwater extraction at any cost for irrigation without jeopardizing food security. In this regard, the Government of Punjab introduced the "Punjab Preservation of Subsoil Water Act, 2009". The present paper examines the impact of the "Preservation of Sub Soil Water Act, 2009" on pre- and post-water levels in Punjab using the difference-in-difference (DiD) approach. The state has witnessed a severe fall of 0.50 m per year and 0.43 m per year for the post-monsoon and pre-monsoon season, respectively. Only 2.62 per cent of wells were in the range of 20–40 m depth in the state in 1996, which increased to 42 per cent and 67 per cent in 2018 for the pre-monsoon period, and post monsoon period respectively, depicting an increase of 25 times. The groundwater depth in high rice-growing(treated) districts declined by 1.53 and 1.39 m than the low rice-growing (control) districts in the pre-monsoon and post-monsoon periods respectively post the enactment of PPSW Act, 2009. A groundwater governance framework is urgently needed to manage the existing and future challenges connected with the groundwater resource.

**Keywords:** groundwater; over-exploitation; difference-in-difference

## 1. Introduction

Groundwater plays an important role in sustaining life on earth as it fulfils the thirst for water of about 1.5 to 2.8 billion people, nearly half of the world's population and major source of irrigation in agricultural economies. No doubt, plenty of benefits have been accrued by the use of this resource like food, feed and fodder production for the ever-increasing population, economic development and rural poverty reduction. Nevertheless, its over-extraction has led to aquifer depletion in many parts of the world [1]. The stories of global groundwater regime when percolating to regional level presents that draft exceeds recharge at most of the places in the world. Many of the world's intense agricultural production regions, like California in the USA [2] and the North China plain [3] are currently experiencing water scarcity. Irrigation has made the problem worse causing detrimental effects on the environment (such as excessive groundwater depletion and increased surface runoff) [4,5]. Groundwater has been depleted in many arid and semi-arid regions due to the long-term overexploitation of groundwater [6,7]. Increased water costs, saltwater intrusion and land subsidence are some of the harmful effects of groundwater depletion [8–10]. Preventing groundwater depletion requires carefully regulated aquifer recharge [11,12]. To achieve the objectives of food security and sustainable water usage,

it is furthermore crucial to match the spatial pattern of irrigation demand and available water resources [13–15].

Looking at the picture of groundwater-using countries, the withdrawal as a percent of total renewable groundwater resources accounts for 109, 108, 954, 350 and 800 per cent in Pakistan, Iran, Saudi Arabia, Egypt and Libya as estimated by FAO's AQUASTAT database. India tops the list with a 28.9 per cent share of global withdrawals, followed by the USA (16.7%), Pakistan (9.1%), China (8.1%) and Iran (8.1%), respectively; making 71 per cent with these top five abstractors (http://www.fao.org/nr/water/aquastat/main/index.stm, accessed on 20 December 2022).

Indian agriculture is heavily dependent on access to groundwater [16]. Groundwater has been made available primarily through a regime of either free or a highly-subsidized provision of electricity for extracting the groundwater [17,18]. Unsurprisingly, this energy pricing regime has resulted in an unprecedented use of electric pumps to extract groundwater and cultivate water-intensive crops, especially in Punjab, which has led to water tables declining steadily [19–21]. Undoubtedly, Punjab was a pioneer in India's agricultural revolution and relieved India from importing food grains from foreign countries, allowing it to become a food-secure nation. Punjab's agriculture is crucial for India's food security; currently, 26 percent of the paddy and 38 percent of the wheat are procured by the Food Corporation of India [22]. The paddy–wheat system has largely been responsible for rapidly declining groundwater levels in Punjab. A comprehensive set of documents has been published by [16,23–31] on groundwater depletion in Punjab. The state has been the country's leading food producer on the cost of excessive groundwater extraction. The present groundwater development in the state is 166 percent, and 79 percent of blocks are overexploited [32]. Clean water, which comes from fresh or groundwater, is important in a variety of economic aspects. Geography and ecology have influenced a lot of the physical–chemical and biological properties of water [33].

Between June and October, when paddy is cultivated across Punjab, groundwater pumping intensity, and hence, electricity consumption, is highest [34]. Although the paddy and monsoon seasons coincide, the amount of rainfall that Punjab receives (average annual rainfall of 635.9 mm) is insufficient, and supplemental irrigation is required [35]. Because the surface water is incapable of satisfying agricultural demand, the groundwater is under increasing pressure. Moreover, the aquifers also become saturated by infiltration which consequently affects groundwater recharge in different ways. The groundwater variation is influenced by the time lag between two rainfall sessions or by the vagaries of the monsoon [36]. The groundwater level in around 85 per cent of Punjab's area has decreased from 1984 to 2016; while in the remaining 15 per cent, it has increased for the pre-monsoon period. The extent of groundwater level rise or decline varied from place to place. A decline in water level of more than 15 m occurred in around 31 per cent of the estimated area in the districts of Barnala, Bathinda, Hoshiarpur, Jalandhar, Ludhiana, Moga, Patiala and Sangrur. In most of the state's territory, the water level has declined. The average annual rate of water level decline is around 0.37 m/year for the pre-monsoon period from 1984 to 2016. On the other hand, the state's average annual rate of water level rise was approximately 0.19 m/year [37]. Irrigation has been the main reason for the high depletion of water levels [38].

As discussed above, the groundwater levels steadily declined in Punjab, the authorities are concerned about reducing extraction at any cost for irrigation without jeopardizing food security. In this concern, the Government of Punjab introduced the "Punjab Preservation of Subsoil Water Act, 2009". The Act forbids the sowing of paddy nurseries before May 10 and transplanting it into the main field before 10 June or any other period specified by the government. It was projected that the postponing of the paddy crop until 10 June can check the decrease in water level by 30 cm and save 276 million kWh power subsidy worth Rs.1220 million [35]. The goal of this analysis was to examine the impact of PPSW Act, 2009 in arresting the groundwater decline in Punjab.

## 2. Materials and Method

### 2.1. Study Period and Type of Data

In India, groundwater is used to irrigate around 45 per cent of the country's total cultivated land. The country's green revolution was brought on by groundwater [39]. Punjab was the locus of the green revolution. In periods of inadequate rainfall, groundwater irrigation has guaranteed food security and enabled a significant rise in agricultural production. Nevertheless, this development pattern is not feasible. Punjab, being an agrarian state with a major area under rice–wheat rotation, is heavily dependent on its groundwater resources. Due to the existing cropping pattern and efforts to enhance food grain output, the irrigation system is under tremendous stress due to limited surface water, which is insufficient to meet requirements resulting in stress on groundwater. Although Punjab has gained tremendously from agricultural growth and prosperity, it has come at a significant cost. The groundwater resources have been declining at an average rate of 53 cm/year for the last two decades. To arrest this declining rate, the Government of Punjab had implemented a sub soil preservation act in 2008 according to which farmers were not allowed to transplant paddy before 10 June. This act become a law in 2009. However, due to the shortening of the window period between paddy harvesting and wheat sowing, incidents of stubble burning increased in the state. Therefore, to understand the consequence, it is imperative to quantify the impact of sub soil preservation act on groundwater resources.

Time series data from 1999 to 2018 for 20 districts of the state were pooled to construct a panel data. Furthermore, the time series was divided into two periods, i.e., 1999–2008 (pre-Act) and 2009–2018 (post-Act) depicting 10 years before and 10 years after the PPSW Act 2009 to assess the impact of the Act. The period 1999–2018 was chosen for analysis as the data for these years and for 20 districts was consistently available for all the variables to be used in the DiD model.

Presently, the state has 22 districts, but due to the non-availability of time series data on the two newly made districts, viz., Pathankot and Fazilka, these were merged with their parent districts, Gurdaspur and Ferozepur, respectively [29]. The Central Ground Water Board and Indian Meteorological Department obtained groundwater levels and rainfall data. The data on population density were obtained from the Population Statistics of Punjab published by the Economic and Statistical Organisation, Punjab. Time series data for population density data were not available as a census occurs every 10 years. Census data were unavailable during the study period, hence, interpolation and extrapolation were carried out to make a time series. In addition, we relied on various issues of the Statistical Abstract of Punjab for the data on canal and tube–well irrigated areas and cropping patterns.

### 2.2. Hydrogeology of Punjab

Based on hydrologically and agro-climatologically, Punjab can be broadly divided into three major zones—Kandi, Central and Southwest zones (Figure 1).

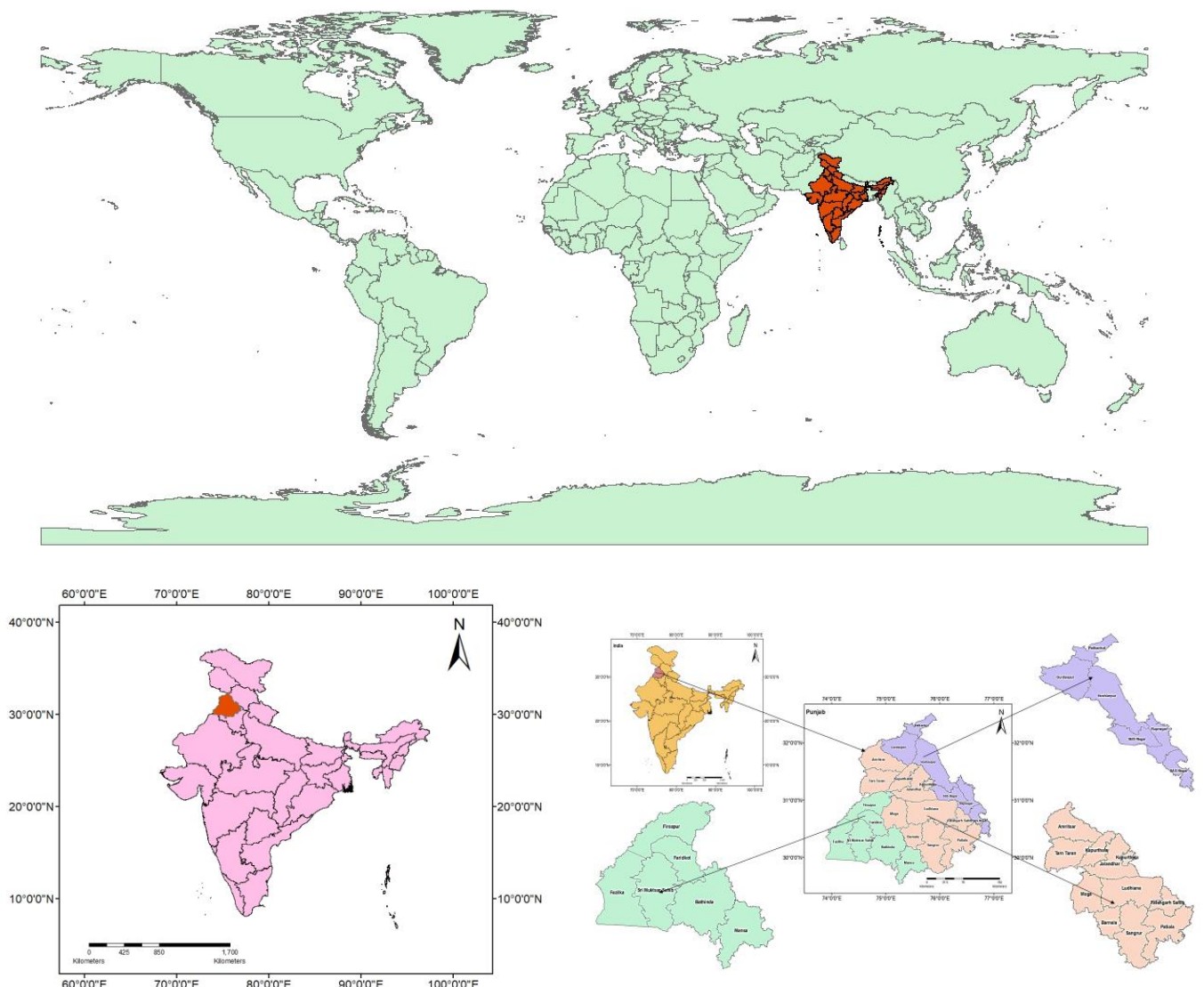

**Figure 1.** Pictorial representation of the study area.

**Kandi zone**: The Kandi zone covers 19 per cent of the geographical area of the state. Due to the steep slope (up to 36 per cent) and high rainfall (1100 mm), about 4.5 lakh ha area (nine per cent of the state's area) of this zone is severely affected by soil and water erosion. The sub-surface aquifers of this zone are alluvial in nature and are made up of a complex heterogeneous mass of silts, clays, gravels, fine and coarse sands. Hydraulic conductivity ranges between 5 and 10 m/day while, the specific yield varies from 0.08 and 0.17 [40]. This zone includes Gurdaspur, Hoshiarpur, SAS Nagar, Rupnagar and SBS Nagar districts with an average annual rainfall of 950 mm.

**Central zone**: The Central zone comprises 47 per cent of the geographical area of the state with Ludhiana, Sangrur, Jalandhar, Patiala, F. Sahib, Amritsar, Tarn Taran, Moga, Kapurthala and Barnala districts with an average annual rainfall of 650 mm. The groundwater is suitable for irrigation and the main cropping system is rice wheat. The aquifers of this zone are alluvial, unconfined and are always covered by a soil crust with a thickness of 0.60–8.0 m. Hydraulic conductivity ranges between 10 and 90 m/day and specific yield ranges between 0.08 and 0.17 [40].

**Southwest zone**: The Southwest zone comprises 34 per cent of the geographical area of the state and includes Bathinda, Mansa, Faridkot, Ferozepur and Muktsar districts with an average annual rainfall of 400 mm [41]. It is commonly known as the cotton belt of the

state; the groundwater is brackish, 70 per cent of the area is canal irrigated, and the problem of residual alkalinity is more serious than salinity. In this zone, subsurface geological alluvium comprises of different layers of clay, clay combined with sand, slit, gravels and pebbles. The hydraulic conductivity value ranges from 4 to 25 m/day and the specific yield varies from 0.05 to 0.16 [40].

The groundwater flows from northeast to southwest. In the Kandi zone, the hydraulic gradient is steep, ranging from 3.30 m to 5.0 m/km, whereas in the central zone, it is 0.33 m/km. Since groundwater elevation contours are broadly dispersed, groundwater flow is slow in the Southwest zone; however, a steep gradient has been seen around Bathinda which may be caused by the low values of lateral hydraulic conductivity and predominant clayey formation [42].

### 2.3. Model Specification and Description

The study employed a difference-in-difference (DiD) approach to study the impact of the PPSW, Act 2009. DiD is used to assess the impact of a specific intervention by comparing the changes in outcomes over time between a group exposed to the intervention called the treated group, and a group that is not called the control group. Before employing DiD, we categorized high and low rice-producing districts. The districts, whose ratio of area under the rice to the total cultivated area surpassed the sample median (0.6) for the year 1999 were considered as high rice growing (treated) districts and the remaining low rice growing (control) districts. In our sample, districts namely Gurdaspur, Amritsar, Tarn Taran, Kapurthala, Ludhiana, Patiala, Sangrur, Barnala, Faridkot and Fatehgarh Sahib were treated districts. Low rice-producing districts i.e., Jalandhar, SBS Nagar, Hoshiarpur, Rupnagar, SAS Nagar, Ferozpur, Muktsar, Moga, Bathinda and Mansa were considered as the controls (Table 1). If the intervention had a significant impact, then the water use in treated districts would have been affected, and hence the water table depth in these districts would have been impacted.

**Table 1.** Districts selected as treated and control in the study frame.

| High Rice Growing Districts (Treated) | | Low Rice Growing Districts (Control) | |
|---|---|---|---|
| Gurdaspur | Patiala | Jalandhar | Ferozepur |
| Amritsar | Sangrur | SBS Nagar | Muktsar |
| Tarn Taran | Barnala | Hoshiarpur | Moga |
| Kapurthala | Faridkot | Rupnagar | Bathinda |
| Ludhiana | F. Sahib | SAS Nagar | Mansa |

The DiD estimated equation is given by:

$$Y_{it} = \beta_0 + \beta_1 Act + \beta_2 Tr + \beta_3 Act \times Tr + \beta_4 Rain + \beta_5 CaIrri + \beta_6 TbIrri + \beta_7 Pd + \beta_8 Cdi + D_i + T_t + \varepsilon_{it} \tag{1}$$

where, $Y_{it}$ is the groundwater level (pre-monsoon or post-monsoon) in district i at time t. The act is a dummy variable for the Act, which takes values 1 for the post-Act period and 0 for the pre-Act period. Tr is a treatment dummy variable which takes the value 1 for treated districts and 0 for control districts. Act×Tr is an interaction variable between Act dummy and treatment dummy which is the parameter of interest. The coefficient $\beta_3$ (interaction term) shows the Act's impact on groundwater levels. Rain is the rainfall level (pre-monsoon or post-monsoon). CaIrri is the ratio of canal irrigated area to total irrigated area and TbIrri is the ratio of tube well irrigated to total irrigated area. Pd is the population density. Cdi represents the Herfindahl crop diversification index. $D_i$ is district fixed effect whereas; $T_t$ is year fixed effects. $\varepsilon_{it}$ is the error term. Unobserved factors common to all districts in a given year were controlled by year and district fixed effects.

Coefficient $\beta_3$ (interaction term) is the parameter of interest. It gives the impact of the Act on the groundwater level. Year-specific common shocks to all districts of the state are

soaked up by the year or time-fixed effects. Time invariant district specific omitted variables that affect the likelihood of treatment are controlled for by including the treatment indicator.

Two variants of DiD model were employed, one for the pre-monsoon period and the other for the post-monsoon period. The estimated equation was regressed for two specifications for each model. Firstly, the interaction term was estimated with district and year fixed effects, i.e., without co-variates.

*Specification 1: without co-variates*

$$\text{Yit} = \beta_0 + \beta_1 \text{Act} + \beta_2 \text{Tr} + \beta_3 \text{Act} \times \text{Tr} + D_i + T_t + \varepsilon_{it} \tag{2}$$

Secondly, the interaction term was estimated using all time-varying covariates such as the ratio of canal irrigated area to total irrigated area, the ratio of tube well-irrigated area to total irrigated area, population density and crop diversification index along with district and year fixed effects, i.e., with co-variates.

*Specification 2: with co-variates*

$$Y_{it} = \beta_0 + \beta_1 \text{Act} + \beta_2 \text{Tr} + \beta_3 \text{Act} \times \text{Tr} + \beta_4 \text{Rain} + \beta_5 \text{CaIrri} + \beta_6 \text{TbIrri} + \beta_7 \text{Pd} + \beta_8 \text{Cdi} + D_i + T_t + \varepsilon_{it} \tag{3}$$

### 2.4. Estimation Procedure

The stationarity of the time series was tested by applying the unit root test. The unit root test exhibits whether the data are stationary or non-stationary and avoids spurious regression. The Levin–Lin–Chu and Harris–Tzavalis unit-root tests were employed, and each variable was tested for the unit root. The panel data estimation procedure uses two approaches (random effect and fixed effect). We tried both, and then the Hausman test was used to choose the better-suited model. The Hausman test rejected the null hypothesis of the random effect model. Hence, the fixed effect model was applied.

We tested the key assumption of DiD estimation called the parallel trend i.e., the outcome in the treatment and control group would follow the same time trend in the absence of the treatment and a parallel trend would have existed between the two groups. The parallel trend assumption was tested by both the graphical inspection of the trend and by performing a 'falsification test or placebo test'. The parallel trend assumption was satisfied by both methods.

The crop diversification was measured using Herfindahl–Hirschman Index (H.H.I.), which measures the degree of concentration in crop type was calculated using

$$\text{H.H.I.} = \Sigma \left( \frac{\text{Area}_i}{\text{Total Cropped Area}} \right)^2 \tag{4}$$

where $\text{Area}_i$ is the area under the 'ith' crop. Therefore, C.D.I. was worked out to measure the extent of diversification by subtracting the HHI from 1. The zero value of C.D.I. indicates specialization and approach towards one specifies an increase in diversification.

### 3. Results and Discussion

### 3.1. Pre-Monsoon, Post-Monsoon and Overall Scenario of Groundwater in Punjab

There are many challenges associated with groundwater in the state of Punjab. A large part of the state is facing declining groundwater levels [43] due to the over-exploitation of the water resources, while there are increasing cases of groundwater pollution due to various human activities.

During the pre-monsoon period, i.e., the month of June, the water table in 1996 was at a depth of 8.6 and 8.8 m in the Kandi and Central zones, respectively, while in the Southwest zone, it was at a depth of 5.2 m (Table 2). By June 2018, the water table in these zones reached 14.7, 21.8 and 10.4 m depth, respectively, nearly twice or thrice in two decades.

A more rigorous trend was seen in the post-monsoon period wherein the water table declined more than twice for the Kandi and Central zones and thrice in the Southwest zone. The reasons are very apparent increase in paddy area and number of tube wells, and a

decrease in rainfall. The water table in the Central zone has been declining consistently, from 8.8 m in 1996 to 12.7, 19.0 and 21.8 m in 2005, 2015 and 2018, respectively. The water table depth in the Central zone (also called as sweet water zone) has declined 2.5 times in the last 22 years owing to the maximum pumping out of groundwater for irrigation in this zone which is apparent from the increase in electricity-operated tube wells.

**Table 2.** Water Table Depth in Punjab, Pre- and Post-Monsoon, Zone-Wise, 1996–2018 (in m).

| Years | Kandi Zone (31°53′ N, 75°90′ E) | | Central Zone (30°84′ N, 75°59′ E) | | South West Zone (30°34′ N, 74°76′ E) | | Punjab (30°84′ N, 75°41′ E) | |
|---|---|---|---|---|---|---|---|---|
| | June | October | June | October | June | October | June | October |
| 1996 | 8.34 | 6.83 | 8.84 | 8.96 | 5.23 | 5.10 | 7.57 | 6.96 |
| 2000 | 8.05 | 7.57 | 8.95 | 9.87 | 5.03 | 5.26 | 7.34 | 7.57 |
| 2005 | 10.76 | 9.51 | 12.67 | 14.06 | 6.71 | 6.39 | 10.05 | 9.98 |
| 2010 | 10.68 | 9.91 | 16.57 | 17.09 | 8.26 | 7.78 | 11.84 | 11.59 |
| 2011 | 11.344 | 9.12 | 16.47 | 18.52 | 8.03 | 7.84 | 11.95 | 11.26 |
| 2012 | 11.86 | 10.04 | 17.59 | 19.20 | 7.29 | 8.82 | 11.02 | 13.68 |
| 2013 | 12.73 | 11.80 | 18.78 | 20.61 | 8.47 | 8.55 | 13.33 | 13.32 |
| 2014 | 12.21 | 12.05 | 18.24 | 21.42 | 8.57 | 9.12 | 12.67 | 14.19 |
| 2015 | 11.91 | 11.97 | 19.02 | 22.31 | 8.91 | 9.05 | 13.28 | 14.44 |
| 2016 | 12.73 | 14.24 | 19.86 | 23.30 | 8.73 | 9.96 | 13.44 | 15.83 |
| 2017 | 13.57 | 13.36 | 21.37 | 24.13 | 9.69 | 10.26 | 14.87 | 15.92 |
| 2018 | 14.72 | 14.36 | 21.89 | 24.86 | 10.36 | 11.09 | 15.66 | 16.77 |
| Average depth (mbgl) | 10.93 [A] | 10.53 [A] | 16.00 [B] | 16.15 [B] | 7.41 [C] | 7.48 [C] | 11.45 [D] | 11.39 [D] |
| Zone average depth | 10.73 [a] | | 16.08 [b] | | 7.44 [c] | | | |

Source: Central Ground Water Board. Notes: Zones are statistically different from each other. Kandi zone with mean water table depth of 10.73 m is different from the Central zone with a depth of 16.08 m and from the Southwest zone, with a depth of 7.44 m; shown in the table with a, b, and c, respectively; using the Tukey–Kramer multiple comparison test for LS means, F-statistics (F value = 35.94 *p*-value < 0.001) is highly significant for zones. LS-means are not significantly different for June and October water table depths, shown in the table with A, B, C and D, respectively.

Punjab state has witnessed a serious decline of 0.50 m per year for the post-monsoon season, whereas the decline for the pre-monsoon season was 0.43 m per year (Figure 2) from 1996 to 2018. This was due to increase in the area under rice from 2.18 m ha (52.06 per cent to NSA) in 1996, to 3.06 m ha (74.43 per cent to NSA) in 2018 [44]. Other reasons include access to free power for irrigation leading to increased tube wells, assured market for paddy and wheat, deficit rainfall, etc [25,45,46].

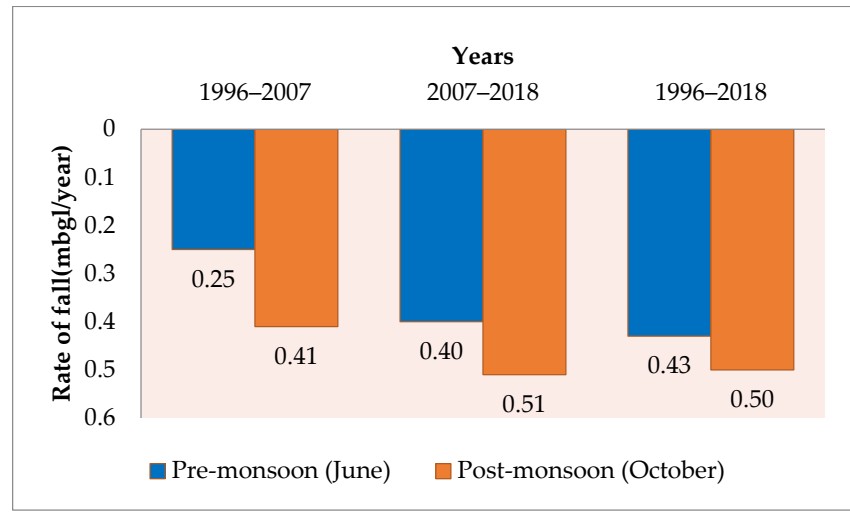

**Figure 2.** Rate of fall in the water table in Punjab (in m/year), 1996–2018.

Pre-monsoon and post-monsoon periods both showed a significant increase in the proportion of wells with a water table depth between 20 and 40 m (Figures 3 and 4). For the pre-monsoon period, in the year 1996, only 2.62 per cent of the wells were between 20 and 40 m depth; however, in the year 2018, the proportion rose to 41.88 per cent, exhibiting a 16-fold growth. Similarly, for the post-monsoon period, the water table depth declined by 25 times, from 2.65 per cent in 1996 to 66.67 per cent in 2018. A major portion of the state registered a fall in the area 10–20 m range from 42.15 per cent to 17.05 per cent for the pre-monsoon period; whereas the post-monsoon period experienced a fall from 39.50 per cent to 19.90 percent, over the duration of 22 years (1996–2018). The wells under water table depths completely changed from the non-critical stage to the critical stage. The percentage of wells in the non-critical stage (water table depth of less than 10 m) was 72.67 per cent in 1996 which fell to 28.37 per cent in 2018 for the pre-monsoon period.

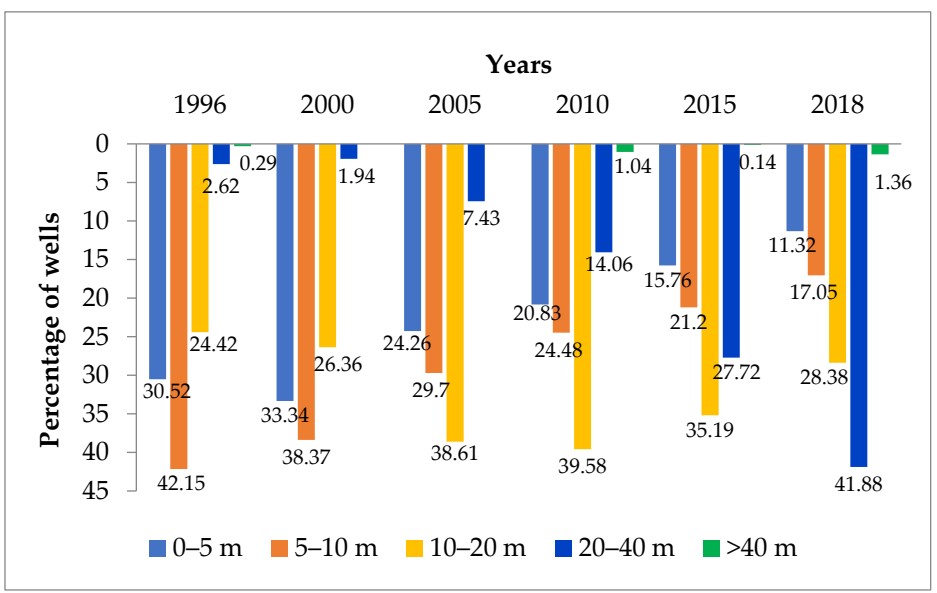

**Figure 3.** Percentage of wells under different water table depths in Punjab, pre-monsoon 1996 to 2018.

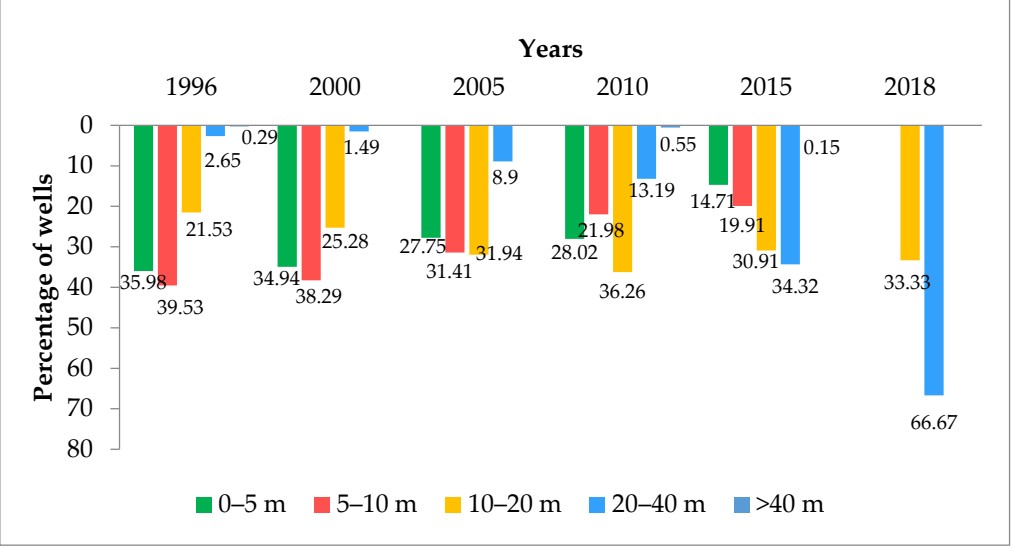

**Figure 4.** Percentage of wells under different water table depths in Punjab, post-monsoon 1996 to 2018. Notes: There is significant effect of years on depth during pre-monsoon period with $\chi^2$ = 1288.89 ($p$-value < 0.001); Likewise, there is also a significant effect of years on depth during post monsoon period with $\chi^2$ = 1373.63 ($p$-value < 0.001).

### 3.2. Estimates of Groundwater Depletion in Punjab

Groundwater depletion was calculated as the decline in the water table multiplied by soil porosity and region area. The estimates are based on the soil porosity of 0.20. The total depletion of water during 1996–2018 was estimated at 93.54 km$^3$ out of which 64.6 km$^3$ was from the Central zone, where paddy area and tube wells are the maximum (Table 3). In the Kandi and Southwest zones, the total depletion during 1996–2018 was 13.61 and 15.33 km$^3$, respectively. The depletion during 1996–2007 for Southwest zone was 6.90 km$^3$, whereas the total depletion through the last 11 years (2007–2018) was more, at 8.43 km$^3$. The water depletion was mostly high during 2007–2018, estimated at 8.59, 34.02 and 8.43 km$^3$ in the Kandi, Central and Southwest zones of Punjab, respectively. The average annual groundwater depletion in the Central zone was 2.78 km$^3$ during 1996–2007, which increased to 3.09 km$^3$ during 2007–2018; for Punjab, it was 1.28 and 1.55 km$^3$ for the respective periods. The average per year water depletion for Punjab was estimated at 1.42 km$^3$, out of which 2.94 km$^3$ was from the Central zone, during 1996–2018.

**Table 3.** Estimates of Total Depletion of Groundwater (in km$^3$) in Punjab, June 1996–June 2018.

| Zone | 1996–2007 | | 2007–2018 | | 1996–2018 | |
|---|---|---|---|---|---|---|
| | Total | Average/Year | Total | Average/Year | Total | Average/Year |
| Kandi Zone | 5.009 | 0.455 | 8.590 | 0.780 | 13.590 | 0.617 |
| Central Zone | 30.604 | 2.782 | 34.021 | 3.092 | 64.626 | 2.937 |
| South West Zone | 6.903 | 0.627 | 8.427 | 0.766 | 15.331 | 0.696 |
| Punjab | 42.516 | 1.288 | 51.038 | 1.546 | 93.547 | 1.416 |

Groundwater recharge declined in all the three zones as the area under rice cultivation increased. With an increase in area under rice cultivation from 52.06 per cent in 1996 to 74.43 per cent of area sown in 2018; the water table depth declined from 8.06 m to 19.11 m, indicating a decline of 11.05 m from 1996 through 2018 in Punjab (Table 4).

In the Kandi zone, the rice area increased from 48.63 per cent to 57.42 per cent of the area sown from 1996 to 2018, respectively, which led to a decline in water table depth from 6.83 m in 1996 to 14.36 m in 2018. In the Central zone, the area under rice increased from 68.47 per cent to 85.84 per cent of the area in the same period. As a result, the decline in water table depth was highest in the Central zone by 15.9 m as compared to the other zones. A maximum decline was found in Sangrur, Patiala, Moga and Barnala.

The water table declined from 5.1 m to 11.1 m during the same period in the Southwest zone. Interestingly, the increase in rice area was highest in the same zone from 366 thousand ha in 1996 to 849 thousand ha in 2018. In the Southwest zone, a large proportion of the cultivated area was traditionally under cotton cultivation, but decreasing yield and price fluctuations, insect–pest attacks and climatic variations, on the one hand, and assured MSP, stable yield and well-established market infrastructure for the rice crop on the other hand, has recently caused a large shift in the area from cotton to paddy [23]. This resulted in a decline in water levels in Bathinda, Mansa, Faridkot and Ferozepur.

Over time, an increase in the rice area has substantially enhanced groundwater use during the monsoon season, causing inadequate recharge in the post-monsoon season. Thus, the change in cropping pattern towards rice is primarily responsible for decline in the water table depth in Punjab [47–49] and the problem in Central Punjab is more severe. As a result, every year, Punjab's water table has been deepening.

**Table 4.** Zone-Wise Water-Level Depth (Oct-over-Oct) and Paddy Area in Punjab, 1996 to 2018.

| Districts | 1996 | | | 2018 | | |
|---|---|---|---|---|---|---|
| | Water Level (m) | Paddy Area | | Water Level (m) | Paddy Area | |
| | | Area Million Hectares | % to Net Sown Area (NSA) | | Area Million Hectares | % to Net Sown Area (NSA) |
| Gurdaspur (32°03′ N, 75°27′ E) | 4.43 | 0.191 | 65.41 | 7.13 | 0.204 | 79.68 |
| Hoshiarpur (31°32′ N, 75°57′ E) | 8.85 | 0.057 | 26.51 | 17.33 | 0.075 | 36.76 |
| SAS Nagar (30°70′ N, 76°72′) | 6.35 | 0.029 | 61.70 | 22.80 | 0.031 | 40.26 |
| Rupnagar (30°57′ N, 76°32′ E) | 5.51 | 0.037 | 48.05 | 13.99 | 0.040 | 49.38 |
| SBS Nagar (31°09′ N, 76°04′ E) | 10.05 | 0.042 | 41.58 | 19.93 | 0.060 | 62.50 |
| Kandi Zone (31°53′ N, 75°90′ E) | 6.83 | 0.356 | 48.63 | 14.36 | 0.410 | 57.42 |
| Ludhiana (30°55′ N, 75°54′ E) | 10.08 | 0.230 | 76.67 | 21.06 | 0.258 | 86.28 |
| Sangrur (30°12′ N, 75°53′ E) | 7.75 | 0.228 | 70.59 | 34.03 | 0.284 | 90.15 |
| Jalandhar (31°19′ N, 35°18′ E) | 9.92 | 0.112 | 49.12 | 25.54 | 0.171 | 70.37 |
| Patiala (30°2′ N, 76°25′ E) | 10.24 | 0.209 | 73.34 | 30.28 | 0.233 | 90.66 |
| F. Sahib (30°64′ N, 76°39′ E) | 9.46 | 0.080 | 77.67 | 23.24 | 0.086 | 84.31 |
| Amritsar (31°37′ N, 74°55′ E) | 5.57 | 0.154 | 67.84 | 14.74 | 0.180 | 82.19 |
| Tarn Taran (31°28′ N, 74°58′ E) | 8.06 | 0.150 | 68.49 | 18.95 | 0.182 | 83.48 |
| Moga (30°82′ N, 75°17′ E | 9.60 | 0.108 | 55.67 | 25.80 | 0.181 | 93.29 |
| Kapurthala (31°23′ N, 75°25′ E) | 9.03 | 0.102 | 75.55 | 18.76 | 0.118 | 88.72 |
| Barnala (30°38′ N, 75°55′ E) | 10.87 | 0.093 | 73.23 | 34.19 | 0.113 | 91.12 |
| Central Zone (30°84′ N, 75°59′ E) | 8.96 | 1.466 | 68.47 | 24.86 | 1.806 | 85.84 |
| Bathinda (30°11′ N, 75°00′ E) | 7.82 | 0.039 | 13.08 | 16.73 | 0.160 | 54.6 |
| Mansa (29°99′ N, 75°39′ E) | 3.72 | 0.050 | 24.51 | 14.85 | 0.107 | 57.83 |
| Faridkot (30°59′ N, 74°83′ E) | 4.32 | 0.038 | 28.78 | 9.14 | 0.115 | 90.55 |
| Ferozepur (30°55′ N, 74°40′ E) | 4.72 | 0.233 | 50.43 | 10.15 | 0.294 | 62.42 |
| Muktsar (30°30′ N, 74°43′ E) | 3.96 | 0.006 | 2.56 | 3.71 | 0.173 | 77.23 |
| Southwest Zone (30°34′ N, 74°76′ E) | 5.10 | 0.366 | 27.52 | 11.09 | 0.849 | 65.31 |
| Punjab (30°84′ N, 75°41′ E) | 8.06 | 2.188 | 52.06 | 19.11 | 3.065 | 74.43 |

Note: Figures in parentheses indicate coordinates of every place i.e., latitude–longitude.

### 3.3. Decreasing Groundwater Balance of Punjab

The cultivation of rice, a water-intensive crop, and the increase in industrialization and urbanization have led to the rise in reliance on groundwater, which has widened the groundwater demand and supply gap over time. This gap is visible in the decreased availability of groundwater in the state, which decreased from 0.27 bcm in 1997 to −14.58 bcm in 2017. The net annual draft has always been higher than the net annual recharge, thus creating a negative groundwater balance indicating a water deficit in the state [23]. The groundwater balance has decreased from 0.027 m ha m in 1997 to −1.063 m ha m in 2017 (Figure 5). A marginal increase in net annual recharge and a decrease in the net annual draft were observed in 2013. The net annual recharge decreased by 46.18 per cent, while the net annual draft increased by 114.66 per cent over the period. The use of groundwater in excess of recharge has led to a decline in the water table and has put huge pressure on groundwater resources.

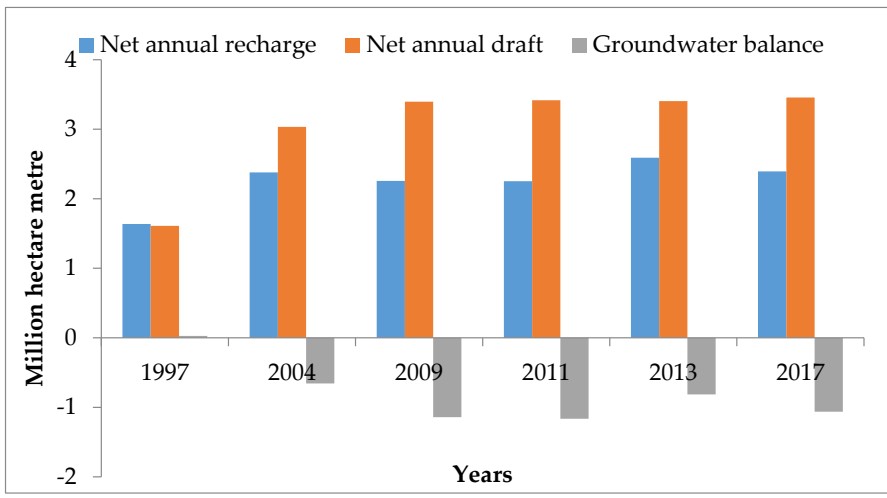

**Figure 5.** Groundwater balance position of Punjab, 1997–2017.

### 3.4. Impact of Sub Soil Water Act, 2009

Before performing the DiD analysis, per cent increase in rice area, number of electricity-operated tube wells and electric tube well density were analysed to see the changes before and after the Act (Table 5). The rice area post the Act showed a decline in high rice-growing districts, whereas it increased in low rice-growing districts. The number of electricity-operated tube wells increased by 26.93 per cent and 40.70 per cent in both the high and low rice-growing districts, post the Act. The density of the tube wells also showed a similar trend. The increase in the post-Act period was 29.41 per cent and 40.84 per cent in high and low rice-growing districts, respectively. The reasons for decline in water table can be attributed to various factors affecting the water table. The decreased recharge over the year has also resulted in more extraction of groundwater for irrigation by the farmers. The same has been depicted by Figures 6 and 7 for pre-monsoon and post-monsoon in the high and low rice-growing districts.

**Table 5.** Percentage increase in rice area and electric tube well density in Punjab.

| Particulars | High Rice Growing Districts | | | | | | Low Rice Growing Districts | | | | | |
|---|---|---|---|---|---|---|---|---|---|---|---|---|
| | Pre-Act (1999–2008) | | | Post-Act (2009–2018) | | | Pre-Act (1999–2008) | | | Post-Act (2009–2018) | | |
| | 1999 | 2008 | % Increase | 2009 | 2018 | % Increase | 1999 | 2008 | % Increase | 2009 | 2018 | % Increase |
| Rice area (million hectares) | 1.59 | 1.67 | 4.95 | 1.72 | 1.78 | 3.84 | 0.95 | 0.96 | 1.79 | 1.04 | 1.31 | 25.55 |
| No. of electricity-operated tube wells (millions) | 0.48 | 0.61 | 25.82 | 0.63 | 0.81 | 26.93 | 0.276 | 0.37 | 34.78 | 0.39 | 0.56 | 40.70 |
| Electric tube well density (million per ha of net sown area) | 0.23 | 0.29 | 28.82 | 0.30 | 0.39 | 29.41 | 0.13 | 0.18 | 34.42 | 0.19 | 0.27 | 40.84 |

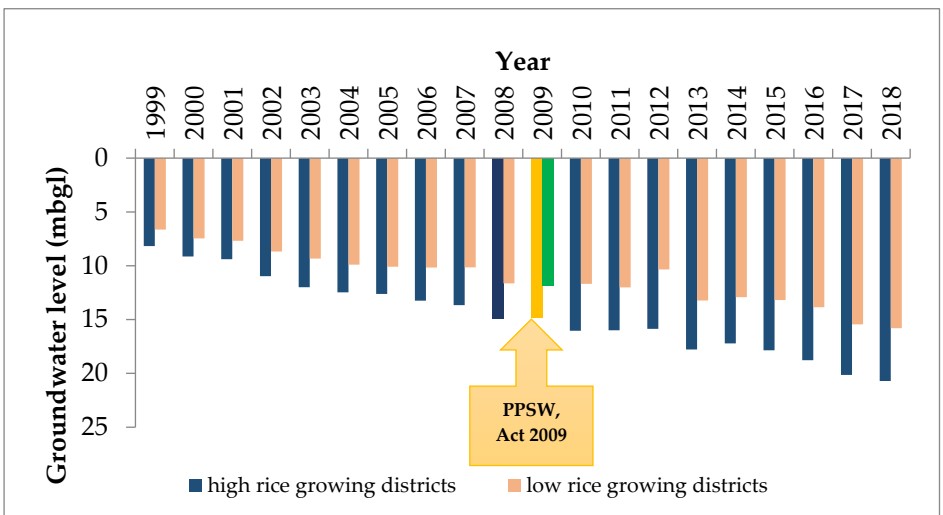

**Figure 6.** Pre- monsoon water levels for high and low rice growing districts of Punjab, pre-Act (1999–2008) and post Act (2009–2018).

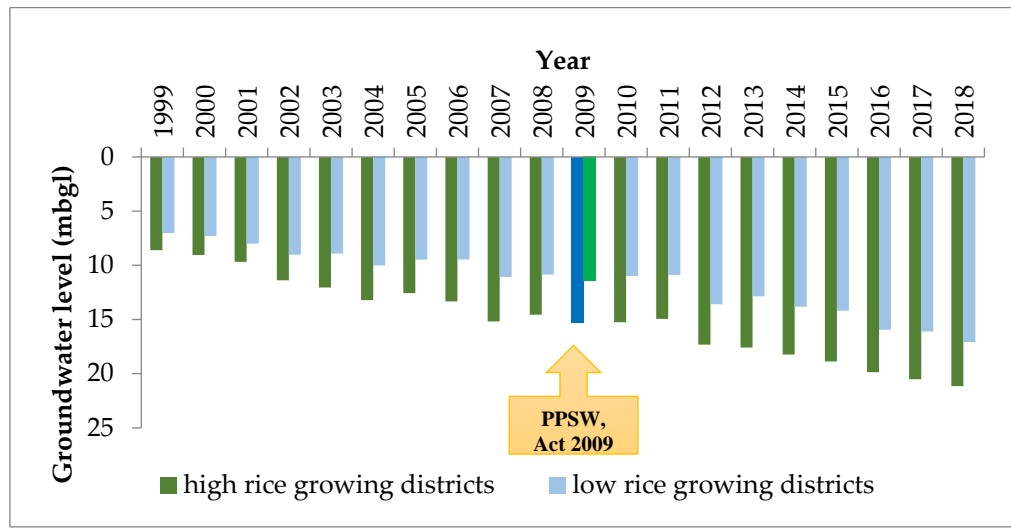

**Figure 7.** Post-monsoon water levels for high and low rice-growing districts of Punjab, pre-Act (1999–2008) and post Act (2009–2018).

The impact of the PPSW Act, 2009 was analysed using the DiD approach to verify the above results. The outcome variable is groundwater depth in mbgl (metres below groundwater level). The sign of the interaction term coefficient (Act*treatment) is interpreted opposite as the water level is measured below the ground. Hence, a positive coefficient of Act*treatment would be interpreted as falling groundwater depth. The Act's effect on groundwater levels for pre-monsoon and post-monsoon periods, respectively, have been presented in Tables 6 and 7.

We used the DiD approach to find the coefficient value of the parameter in two ways, firstly, estimates of interaction term with district and year fixed effects, i.e., without covariates. Secondly, estimates of interaction term with all time-varying covariates such as the ratio of canal irrigated area to total irrigated area, ratio of tube well irrigated area to total irrigated area, population density and crop diversification index along with district and year fixed effects. Overall, the fit of both the models is good as indicated by the $R^2$ which explains around 70 per cent the variation of both the models.

In both the models, even after implementing the policy reform, i.e., the PPSW Act, 2009, the groundwater depth has declined more in treated districts (high rice-growing). The groundwater depth in high rice-growing districts post (2009–2018) the enactment of

PPSW Act 2009 was 1.72 and 1.55 m deeper than the low rice-growing districts in pre-monsoon and post-monsoon groundwater levels models, respectively (Tables 6 and 7). Controlling demographic characteristics and other related variables such as irrigated area, canal availability, cropping pattern and rainfall reduces the decline in groundwater to 1.53 and 1.39 mbgl in pre-monsoon and post-monsoon periods depicting the effect of these on the groundwater regime of the region. In both the specifications i.e., with and without, the PPSW Act, 2009 increases the groundwater depth. The groundwater levels had declined more post the policy reform in treated districts. The coefficients are significant at 1 per cent significance level.

**Table 6.** Impact of Preservation of Sub Soil Water Act, 2009 on Pre-Monsoon Groundwater Levels.

| Particular | Coefficient Value | |
|---|---|---|
| | without Co-Variates | with Co-Variates |
| Act×treatment | 1.72 *** | 1.53 *** |
| | (0.44) | (0.45) |
| Pre-monsoon rain | No | Yes |
| Ratio of canal irrigated area to total irrigated area | No | Yes |
| Ratio of tube well irrigated area to total irrigated area | No | Yes |
| Crop diversification index | No | Yes |
| Population density | No | Yes |
| District fixed effects | Yes | Yes |
| Year fixed effects | Yes | Yes |
| Observations | 400 | 400 |
| $R^2$ | 0.69 | 0.70 |

Note: *** denotes significance at 1 per cent level. The figures in parentheses are standard errors.

**Table 7.** Impact of Preservation of Sub Soil Water Act, 2009 on Post-Monsoon Groundwater Levels.

| Particular | Coefficient Value | |
|---|---|---|
| | without Co-Variates | with Co-Variates |
| Act×treatment | 1.55 *** | 1.39 *** |
| | (0.44) | (0.45) |
| Post monsoon rain | No | Yes |
| Ratio of canal irrigated area to total irrigated area | No | Yes |
| Ratio of tube well irrigated area to total irrigated area | No | Yes |
| Crop diversification index | No | Yes |
| Population density | No | Yes |
| District fixed effects | Yes | Yes |
| Year fixed effects | Yes | Yes |
| Observations | 400 | 400 |
| $R^2$ | 0.71 | 0.72 |

Notes: *** denotes significance at 1 per cent level. The figures in parentheses are standard errors; the values of $R^2$ of without and with co-variates were taken from the Equations (2) and (3), which signifies that the explanatory variables explained 71% and 72% of the total variation, respectively.

The groundwater level has declined at a higher rate in high rice growing districts in comparison to low rice growing districts even after the Act's enactment in the state. It was observed that this Act, 2009 has sensitized people towards groundwater conservation or awareness about the implications of water depletion. Otherwise, the rate of fall would have been even more which got arrested due to this act.

Some evidence shows that this regulation has prevented the rate of groundwater depletion. Using time series data from 1985 to 2011, authors [49] found an annual rise of 0.2 cm in the groundwater table after the implementation of the Act, despite the increase

in the area under rice cultivation [50]. Another study [35] estimated a 30 cm water-saving effect of the Act, but both these studies used different methodologies, which does not account for selection issues.

The results are consistent with Sheetal Sekhri [51] who reported a decline in yearly groundwater levels even after the enactment of the PPSW Act, 2009 in the paddy growing region of Punjab and Haryana and Kishore et al. [21], who found that the PPSW Act, 2009 was unable to control the deepening of the water level in the post-treatment period.

The change in the cropping pattern towards rice is primarily responsible for the degradation of water and soils of Punjab. The high rate of depletion has been excruciatingly in the North China Plain (22.0 mm/year), the high plains of the USA (27.6 mm/year) and north-western India (40 mm/year) [38,52–54]. China represents a similar picture of water scarcity [55,56]. Rice production, which alone consumes about 50% of the freshwater resources in China [57], is threatening its rice production as its groundwater use has been estimated to be 40 percent higher than registered in Aquastat (FAO's AQUASTAT database). In many rice-growing regions, the most significant barrier to rice production is drought stress [55]. Irrigated rice producers will be forced to diversify their method of cropping by cultivating aerobic rice, rainfed rice, maize and other dryland crops as scarcity of water threaten rice production in many regions of the world [58]. Over the past 20 years or more, the water tables in the hard rock aquifers of the southern Indian peninsula and Indo–Gangetic Plains of South Asia have been declining at the rate of one meter per year [59–62].

Falling water tables are also a serious problem in Australia, Spain and the USA. Several countries have introduced legislation to control groundwater development and impose restrictions on actions that can jeopardise the quality and availability of groundwater. Some of the water acts or laws had positive impacts while others have not turned out to be a promising solution. Australia's Federal Water Act 2007 was introduced with the aim to bring water allocations in the Murray–Darling Basin (MDB) back to sustainable levels and to integrate planning and decision-making at the basin level. The Act did this by creating the Murray–Darling Basin Authority (MDBA) and mandating it to create a basin plan by 2019 that set sustainable limits to water diversions in the basin. However, a lack of consistent establishment of boundaries i.e., defining the boundaries of the area so that they align with the hydrogeological boundaries of the aquifer, led to difficulties in managing this act [63].

This pattern shows rising competition and conflict among groundwater users as well as the growing danger of groundwater contamination. The increased integration of legal laws on water resources is the result of the recognition that adverse effects on groundwater may also impact surface water. Similarly, The Water Framework Directive, adopted in 2000 in the European Union, offers an appropriate framework to address the scarcity of water, but an impact study reveals that many questions remain unaddressed. Similar results have been reported for the impact assessment of Sustainable Groundwater Management Act (SGMA), 2014, legislation adopted by the state of California on groundwater extraction patterns in the Sacramento River Hydrologic Region using the DiD regression model. The SGMA grants local governments the power to manage groundwater sustainably and permits limited governmental action to safeguard groundwater resources. The groundwater trends in the treatment and control basins were not significantly different by the study. The study found no evidence that the implementation of groundwater regulation had a measurable difference in groundwater extraction in California, USA [64].

However, the results of legislation were encouraging in the Ogallala aquifer in the central United States. Initially, the problem of groundwater depletion was so severe in that the only solution left was to let the region depopulate and leave it to a grazing ground for buffalo. However, in 2012, a law was passed that makes it possible to construct more choices for leasing water, retiring water rights based on incentives, allocating water over a longer period, and implementing Local Enhanced Management Areas. The most noted achievement has been in the collective action which implemented a Local Enhanced

Management Area (LEMA), which proved that reduced water pumping resulted in little-to-no groundwater degradation while maintaining the net income. The fact that irrigators who have taken part in LEMA or other conservation initiatives have conserved much more water than their targets is even more encouraging.

The literature shows that ordinances passed involving the major stakeholders, i.e., end users in the conservation process have been more successful than others. For example, the Groundwater Conservation Ordinance in Niseko, Japan, 2011, established by the municipal government, in which the water users had the accountability of replenishing groundwater. In Japan, local governments have been autonomously safeguarding their groundwater resources through local ordinances. Groundwater ordinances only exist at the local level, and there is no fundamental groundwater legislation that the government has established.

The possible reasons for this groundwater overdraft could be the following: a large number of irrigations after delayed transplanting of the paddy, free electricity to farmers which allows for more groundwater pumping, an increase in paddy area, replacement of centrifugal pumps into submersible pumps and deepening of already existed submersible ones which can fetch water from deeper layers or aquifers [21,50].

## 4. Conclusions and Policy Implications

The groundwater system of Punjab provides food security to India, and the current groundwater over-exploitation crisis in this region is a pressing concern, as trends in groundwater are not very encouraging from a sustainability point of view. The cropping pattern of paddy and wheat crops has adversely affected the water resources. Due to the continuous rotation of the paddy and wheat cropping pattern, the water and soil of Punjab have been degraded and depleted. The rate at which the groundwater resources are exhausted, without a recharge, has put huge pressure on the state's groundwater resources. Groundwater legislations and laws passed in different countries as innovations to reverse groundwater depletion have achieved limited success due to the open access nature of this resource and the political economy of water management, especially in agriculture. The same is exactly true for the Punjab state. PPSW Act, 2009 though, put a halt on declining water tables in the state. However, the increase in paddy area from 2.7 million to 3 million, irrigation pumps set from 1 million to 1.4 million, accompanied by the declining average rainfall from 650 mm to 450 mm post the Act, led to a decline in groundwater level in high rice-growing districts. The rate of decline became arrested, otherwise it would have been even higher without the PPSW Act, 2009. The study suggests that delaying the transplanting of the paddy, combined with other management strategies such as laser land leveling, direct seeded rice, and sensor-based application of irrigation in rice, will lead to more water saving in rice and hence will check the decline rate. Therefore, water saving technologies and practices should be promoted on a large scale by incentivizing farmers with attractive packages such as the one introduced as a pilot project in the Jalandhar and Hoshiarpur districts of Punjab state by the department of power i.e., "Paani Bachao Paise Kamao", which could not reap the benefits owing to provisions of tariff-free electric power to irrigation sector in the state and some other institutional reasons. Delaying the transplanting of paddy necessitates more research into early maturing cultivars in order to preserve the sowing time of wheat. The high rice-producing irrigated regions should learn a lesson from Punjab's water scarcity, and therefore regulate the use of groundwater through groundwater governance which will not completely control but can halt the pace of declining water levels and check the further depletion of this valuable resource. The depletion of groundwater resources is among the most severe problems of Punjab agriculture, which needs more corrective measures. That could be possible with the effective implementation of this Act. Moreover, the government need to amend the PPSW Act, 2009 in a systemic manner which considers each agro-climatic zone, rather than one for the state as a whole. Additionally, a major emphasis should be on the addition of new crops in the cropping patterns, limiting the groundwater draft or increasing the recharge of groundwater, harvesting rainwater and using smart technological methods to

improve irrigation efficiency and the use of treated wastewater to put a halt on the rate of groundwater depletion.

**Author Contributions:** Conceptualization, Y.S. and B.K.S.; methodology, Y.S., B.K.S., S.K. (Samanpreet Kaur), M.K.S. and A.K.M.; software, Y.S., B.K.S., S.K. (Sunny Kumar) and S.K. (Samanpreet Kaur); validation, Y.S., B.K.S., S.K. (Sunny Kumar) and S.K. (Samanpreet Kaur); formal analysis, Y.S., B.K.S., S.K. (Sunny Kumar) and S.K. (Samanpreet Kaur); investigation, Y.S., B.K.S., S.K. (Sunny Kumar) and S.K. (Samanpreet Kaur); writing, Y.S., B.K.S., S.K. (Sunny Kumar), S.K. (Samanpreet Kaur), M.K.S., A.K.M. and S.M.; writing Y.S., B.K.S., S.K. (Sunny Kumar), S.K. (Samanpreet Kaur), M.K.S., A.K.M. and S.M. All authors have read and agreed to the published version of the manuscript.

**Funding:** This research received no external funding.

**Institutional Review Board Statement:** Not applicable.

**Informed Consent Statement:** Not applicable.

**Data Availability Statement:** Data will be made available on email request.

**Conflicts of Interest:** The authors declare no conflict of interest.

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
