# Peer review of "Pre and Post Water Level Behaviour in Punjab: Impact Analysis with DiD Approach"

_sustainability, doi:10.3390/su15032426_

Round 1

Reviewer 1 Report

The article entitled "Pre and Post Water Level Behaviour in Punjab: Impact Analysis 2 with DiD Approach" is well written but contains many errors. First of all, it concerns only a small region in India. The obtained results are of local importance and are not compared with other regions of the world. I gave my comments in the text of the manuscript. The article after major corrections should be re-reviewed.

Author Response

Respected Sir/Madam,

Please find enclosed the authors response to the reviewer1's comments.

Thanks and Regards

Dr Baljinder Kaur Sidana

Reviewer 2 Report

1) Equation at line 59 needs to be numbered. 

2) Table 1,  coordinates for every place required.  

3)Unit S.I. for figure 1. 

4) Table 3, coordinates for every place needed. 

5)Table 4-R2 gets from which equation? Need to explain. 

6)Line 50 can be continued with:- 

Clean water which is come from fresh or ground water is important in a variety of economic aspects. Geography and ecology have influenced a lot of physical-chemical and biological of water (Tajarudin H.A et al., 2019) 

"HAB Tajarudin, MMZ Makhtar, MS Azmi, NI Zainuddin, DHA Ali (2019) Introduction to water and wastewater treatment. Advanced oxidation processes (AOPs) in water and wastewater treatment, 1-29"

Author Response

Respected Sir/Madam,

Please find enclosed the reviewer2's comments.

Thanks and Regards

Dr Baljinder Kaur Sidana

Reviewer 3 Report

The manuscript “Pre and Post-Water Level Behavior in Punjab: Impact analysis with DiD approach” gives insight into the status of the Preservation of subsoil water act, 2009. The manuscript clearly depicts the inability of acts to control groundwater depletion. The structures of the manuscript are in presentable form for the journal, however, some of the corrections are made before the final publication. Please consider the specific comments added and the highlighted parts in the file attached along with the comments.

Comments

1.     In Lines 29-30, the authors specified the groundwater depth in the high rice growing district was deeper than the low rice growing area. Does this specify the water table depth below the ground surface or lower water table depth compared to a low rice-growing district?

2.     Please rewrite the sentence in lines no 57-58.

3.     I suggest using the word decline than fall. Please amend the best word describing the groundwater level reduction.

4.     Please consider the pictorial representation of the study area in Figure 1. Also please remake the map with better quality. If possible, try to indicate all the zones (Kandi, Central, South, Punjab) in Punjab. Also, the scale of the map is not clearly visible. Please consider the map style.

5.     Since the authors are dealing with groundwater, the major missing in the manuscript is the hydrogeological information on Punjab. Since, the author made a depth-wise discussion, the declining water level in 20-30m and increasing water level in 10-20 will be clearer. Please specify whether the region contains a homogenous/ heterogeneous aquifer.

6.     A bit of confusion arose while using multiple words on increase and decrease, please use consistent wording for increase, falling, rose, and decrease in Lines 196-206. Especially in Line no 200 where the author specified the water table depth rose from ….

7.     What do the non-critical stage and critical stage refer to?

8.     Line 297-298, how can the author be so sure of the changes has been made after the act on the people?

9.     Please add the x-axis in Figure 2 and Figure 3. Additionally, legends are missing for the water level higher than 20. Also if possible, please invert the graph to represent the decline in groundwater level.

10.  Please, reformat tables 3, 4, and 5.

Author Response

Respected Sir/Madam,

Please find enclosed the reviewer3's comments.

Thanks and Regards

Dr Baljinder Kaur Sidana

Round 2

Reviewer 1 Report

I accept in present form, but authors have to add: 

in line 149. Fig. Pictorial representation of the study area

in line 197: Tab. 1.  ... (Title of this table)

Authors should number the drawings and tables correctly

Author Response

Respected Sir/Madam

Please find enclosed the revised manuscript (changes are highlighted in yellow).

Thanks and Regards

Baljinder Kaur Sidana

Reviewer 3 Report

All the comments and suggestions have been added to the manuscript. However, please make sure to add high-quality images in Figure 1 and be sure to make necessary changes in the borders and GPS data included in the map.

Author Response

(The authors gave the same response as above.)
